# Predictors of Reactive Agility in Early Puberty: A Multiple Regression Gender-Stratified Study

**DOI:** 10.3390/children9111780

**Published:** 2022-11-19

**Authors:** Vladimir Pavlinovic, Nikola Foretic, Sime Versic, Damir Sekulic, Silvester Liposek

**Affiliations:** 1Faculty of Kinesiology, University of Split, 21000 Split, Croatia; 2Faculty of Mechanical Engineering, University of Maribor, 2000 Maribor, Slovenia

**Keywords:** non-planned agility, pre-planned agility, anthropometry, children

## Abstract

Reactive agility (RAG) is a crucial factor of success in sports, but there are practically no studies dealing with RAG among children. The main aim of this study was to identify predictors of RAG among early pubescent boys and girls. The participants were primary school boys (*n* = 73) and girls (*n* = 59) aged 11–12. The criterion variable was the originally developed “Triangle” test of reactive agility (Triangle-RAG). Predictors included anthropometric/body composition indices (body height, seated height, body mass, and body fat percentage) and motor abilities (10 and 20 m sprint, broad jump, squat jump, countermovement jump, drop jump, and two tests of change of direction speed—CODS (Triangle-CODS, and 20 yards)). The results of the univariate analysis showed that anthropometric/body composition indices were not significantly correlated to TRAG (0–4% of the common variance), while all motor abilities were significantly associated with TRAG (7–43% of the common variance) in both genders. Among boys, 64% of the TRAG variance was explained by multiple regression, with TCODS as the only significant predictor. Among girls, multiple regression explained 59% of the TRAG-variance with TCODS, countermovement jump, and drop jump as significant predictors. Differences in multivariate results between genders can be explained by (i) greater involvement in agility-saturated sports (i.e., basketball, tennis, soccer) in boys, and (ii) advanced maturity status in girls. The lack of association between anthropometric/body built and TRAG was influenced by the short duration of the TRAG (3.54 ± 0.4 s). Our findings suggest that pre-pubescent and early pubescent children should be systematically trained on basic motor abilities to achieve fundamentals for further developing RAG. Since in this study we observed predictors including only athletic abilities and anthropometric/body composition, in future studies, other motor abilities, as well as cognitive, perceptual, and decision-making parameters as potential predictors of RAG in children should be investigated.

## 1. Introduction

Agility is often defined as the ability to quickly and efficiently change the speed and direction of movement [1,2]. It belongs to the domain of speed-explosives abilities that are determined by training status and genetic potential [3]. Agility is a complex motor ability, and in the background of the manifestation of agility are numerous other abilities, such as speed, power, coordination, balance, and even cognitive and perceptual capacities [4]. In general, the existence of two basic types of agility is widely accepted and scientifically proven; (1) change of direction speed or pre-planned, non-reactive agility (CODS) and (2) reactive, non-planned agility (RAG) [5,6]. CODS is evident in situations where the movement pattern is known in advance, while on the other side, in the RAG, agile movement is performed based on some external stimuli and cannot be pre-planned [7,8]. It is generally accepted that different factors influence these two agility manifestations. Briefly, while CODS is mainly determined by morphological and motor parameters, perceptual and cognitive skills are crucial in RAG [3,9].

Both facets of agility are present in sports, and their importance to situational efficiency and performance is confirmed [10,11]. Although it represents one of the most important motor skills in athletes, recently, its importance has been highlighted outside the competitive context [12,13]. In particular, agile movements are presented in professional activities (i.e., military, police), in everyday life, regardless of age, while performing unexpected reactions, overcoming obstacles, and, more specifically, solving perceptual-motor tasks during playing time [14,15,16].

Agility measurements among children are important for one specific reason. In particular, when performing agility tasks, a specific movement technique that differs from sport to sport plays a significant role [3,17]. This technique is developed through systematic, sport-specific training and will play a significant role in the performance of agility when testing adult athletes. Consequently, this will not give a precise picture of the abilities and skills that affect agility, regardless of movement efficiency. For this reason, it is necessary to analyze the predictors of the agile movement in the population of children, especially among the ones who do not engage in agility-saturated sports.

Several studies investigated the predictors of non-reactive, pre-planned agility in children not involved in specific sports [4,18,19]. For example, the study on early pubescent girls identified reactive strength as the most significant predictor, while body composition and anthropometrics had weak-to-medium correlations with reactive agility performance [18]. More recently, a study on early pubescent boys and girls analyzed the influence of balance, jumping, speed capacities, and several morphological variables on three different agility tests [19]. Results highlighted sprinting at 10 m, body mass, and high jumps as the most important predictors [19]. Additionally, a study on early pubescent boys investigated predictors in five different agility performances [4]. Predictors explained between 47% and 62% of the variance, with the two-leg lateral jumps recognized as the single best predictor [4].

Although studies already analyzed predictors of CODS, there is an evident lack of studies exploring the predictors of RAG among children. Meanwhile, RAG is known to be an important determinant of success in agility-saturated sports [7,20,21]. A better understanding of the background of RAG in children will hopefully result in a more accurate orientation of talented children toward agility-saturated sports (i.e., basketball, soccer, handball, tennis). Therefore, the main aim of this study was to determine the association between anthropometric/body composition indices, motor abilities (predictors), and RAG in early pubescent boys and girls. Knowing the differences in fitness status between prepubescent boys and girls, we tried to avoid the potential influence of gender as a covariate of established associations; therefore, a gender-stratified approach was applied. We hypothesized that the studied predictors would be significantly associated with RAG with some gender specifics.

## 2. Materials and Methods

### 2.1. Participants

Primary school boys (*n* = 73) and girls (*n* = 59) aged 11–12 years were involved in this study. In the first phase of the study, a sub-sample consisting of 21 participants was tested on newly developed tests throughout the test–retest procedure in order to evaluate the reliability of the tests (for details on reliability, please see the first part of the Results section). All participants were in good health and were regularly attending physical education classes (PE), while some of them were included in out-of-school sports. The inclusion criteria were: no evident motor aberrations and health-related issues (as indicated by school medical staff), no locomotor injury over a period of two weeks before testing, and regular participation in PE. Exclusion criteria were: recent musculoskeletal disorders, sickness over the previous two weeks, the current prevalence of pain, and/or overall sense of weakness, and three participants were excluded from the study accordingly.

The Ethical Board of the Faculty of Kinesiology University of Split, Split, Croatia, administered approval for the investigation (Ethical board number: 2181-205-02-05-22-0021). The participants were informed of the purpose of the study, and the written consent was signed by their parents or custodians.

### 2.2. Measures and Procedures

Variables in this study included predictors and criteria. The predictors consisted of anthropometric/body-built indices (body height, seated height, body mass, and body fat percentage), motor abilities (10 m sprint—S10M, and 20 m sprint—S20M, broad jump—BJ, squat jump—SJ, countermovement jump—CMJ, and drop jump—DJ, triangle test of change of direction speed—TCODS and 20 Yard shuttle agility test—20Y). The criterion variable was the originally developed “Triangle” test of reactive agility (TRAG).

Body height was measured using a Seca Instruments stadiometer. Body mass and body fat were assessed using a Tanita Pro MC-780U body composition analyzer (Tanita Corp., Tokyo, Japan), which provides a print-out of the measured body mass and calculated body fat. Information about the participants’ gender, age, and body height was inserted into the device, and the participants had to stand barefoot in an upright, stable position. The device provided body mass and used an algorithm incorporating impedance, age, and height, to estimate the percentage of the total body fat.

A Brower timing system (Salt Lake City, UT, USA) was used for the assessment of S10 and S20, which is a commonly used and previously validated system [22]. Two electronic timing gates were placed 1, 11, and 21 m from the starting line. These photocells were mounted 1 m above floor level, which is the maximal height of the manufacturer’s standard tripods. The participants ran as fast as possible for the required distance, with the self-chosen preferred leg placed on the starting marking.

For the 20Y, TCODS, and TRAG agility tests, BlazePod was used (Play Coyotta Ltd., Tel Aviv, Israel).

For TCODS and TRAG, three lighting pods were mounted on 50 cm cones in an equilateral triangle formation—equal sides, equal angles of 60° (Figure 1).

To set the angles, a universal plastic goniometer with a 360° conveyor was used (1° accuracy, European Product). The distance between the cones was set at 4.5 m. In the TCODS test, the participants knew the scenario in advance (first trial: A–B–C, second trial: A–C–B), and after starting, they had to run from one cone to another and touch the lighting pods to turn off the lights. For the TRAG test, the participants did not have advanced knowledge of the scenario. However, all participants were tested by the same templates (first trial: A–C–B, second trial: A–B–A, third trial: A–C–A, fourth trial: A–B–C). The participants were instructed to begin the test with their preferred foot forward placed next to the starting cone. To start the TRAG, students should tap the first lighting pod (cone A), run to the next lighted cone, and touch the designated pod, which triggers the last cone. TRAG scenarios were applied in random order, but all participants were tested in all four scenarios. The testing was arranged in groups of 4–5 participants, which allowed for appropriate rest intervals between the tests and trials. The rest interval was not less than 20 s between trials. 

For the 20Y test, three 50 cm cones, with lighting pods on top, were positioned along a line 4.57 m (5 yd) apart. Students would start with a two-point stance after touching the middle pod to run fast as possible, 4.57 m to the left. The subjects then ran 9.14 m to touch the illuminated cone on the right and finally finished by running back, touching the middle pod.

SJ, CMJ, and DJ testing were performed using the OptoGait system (Microgate, Bolzano, Italy). The software platform allows for the easy storage of all tests carried out and the ability to recall them instantly if necessary. Before the test, the participants were familiarized (the test procedure has been explained to the participants) with the procedure and had three attempts of each test. The students had to use maximal effort to achieve the best possible result. During the broad jump test, the participants stood on the starting line with their legs parallel and feet shoulder-width apart. They were instructed to bend their knees (the degree of flexion was determined by the participant) and bring their arms behind their bodies. A powerful drive was then used to propel them forward.

All measurements were performed on an indoor gymnasium with a wooden floor. Before testing, the participants completed a 10 min warm-up including jogging, skipping, lateral running drills, dynamic stretching, and light jumping. The testing protocol was the same for all participants. All of the tests were performed at the same time of day (9 to 11 a.m.) to prevent variations in the biorhythm and fitness abilities. The participants had one practice trial for familiarization with each test and performed it with self-chosen sports shoes. For the tests measured automatically by the Brower timing system, Optogait, and the Blazepod system, the same examiner assessed all participants.

### 2.3. Statistics

Statistical calculations included several groups of analysis. First of all, a Kolmogorov–Smirnov test was used to check the normality of distribution, and means and standard deviations were calculated for all observed variables. The reliability of the agility tests was checked by calculating Intra Class Coefficients (ICC). Student’s T-test was used to evaluate the differences between the genders. To examine the univariate associations between the variables, the Pearson correlation coefficient was calculated. In the last phase, all significantly correlated variables were included in the multiple regression analysis to identify the predictors of the TRAG, separately for boys and girls. For all the analyses, Statistica 14.0 (TIBCO Software Inc., Palo Alto, CA, USA) was used, with a p-level of 95% in all calculations.

## 3. Results

The TRAG and TCODS tests showed appropriate inter-testing and intra-testing reliability (TRAG: ICC = 0.69 and 0.76, TCODS: 0.77 and 0.80 for inter-testing and intra-testing reliability, respectively).

The descriptive statistics and results of the Kolmogorov–Smirnov test of the normality of distributions for all variables are presented in Appendix A (for boys and girls, respectively).

Table 1 presents the results of univariate correlations for boys. Apart from the small and negligible correlation between BF and TRAG (less than 5% of the common variance), anthropometric/body-built indices were not significantly correlated to TRAG. However, practically all motor variables were significantly correlated to TRAG (6% to 60% of the common variance).

Among girls, anthropometric/body composition indices were not correlated with TRAG, while all motor indices except 20Y were significantly correlated with TRAG (10–39% of the common variance) (Table 2).

When multiple regressions were calculated for boys, 64% of the variance was attributed to the TRAG, with TCODS as a single partial significant regressor (Table 3).

When the multivariate relationship between the predictors and TRAG was calculated for girls, 59% of the variance in the TRAG performance was explained. The significant partial predictors were TCODS, CMJ, and DJ, with better reactive agility in girls who perform better in CODS and jumping tests (Table 4).

## 4. Discussion

This study aimed to identify predictors of reactive agility among early pubescent boys and girls. There are several very important findings. First, anthropometric/body built indices were not correlated with TRAG in the studied children. Second, multivariate analysis evidenced TCODS as the only significant multivariate predictor of TRAG in boys. Meanwhile, in girls, in addition to TCODS, leg power was highlighted as a significant multivariate predictor. Therefore, our initial study hypothesis was confirmed.

### 4.1. Anthropometric/Body Composition Indices and Reactive Agility

Anthropometric/body built indices were already studied as being potential predictors of facets of agility in children, but almost exclusively in relation to pre-planned agility (e.g., CODS), and the findings were not consistent [4,19]. For example, in the study on early pubescent boys, Sekulić et al. found no significant correlation between observed anthropometry indices and five different pre-planned agility tests except for body mass and the Zig-zag test [4]. On the other hand, Pavlinović et al. reported a significant correlation between body mass and body fat with pre-planned agility in both boys and girls [19]. Meanwhile, to the best of our knowledge, this is the first study where anthropometric/body-built indices were observed as predictors of reactive agility. In short, apart from the negligible correlation between body fat and TRAG in boys (less than 5% of the common variance), anthropometric/body-built indices were not associated with TRAG in early pubescent children.

The first reason for the absence of an association between anthropometric/body composition and TRAG can probably be found in the duration of the test applied in our study. Namely, the test duration was very short (approximately 3 s). It, therefore, did not contain a significant energetic component, for which a higher body and fat mass would represent an important factor of influence. Second, it is widely accepted that reactive agility is more a complex ability than CODS, being under the influence not only of conditioning capacities and corresponding anthropometric/body built indices but also cognitive-perceptual abilities (REFS). As a result, simply mathematically/statistically, the percentage of the RAG variance which could be explained by anthropometrics/body built is reduced, resulting in negligible correlations observed herein. It is also important to highlight that our study observed participants (both boys and girls), who were mostly in the pre-peak high velocity (PHV) age. As a result, there was no significant difference between them in anthropometric indices that could affect RAG performance [23,24]. Consequently, we have found no evidence that anthropometric/body-built indices should be observed as significant predictors of RAG in this age group.

### 4.2. Motor Abilities and Reactive Agility

Analyzing the results of univariate correlation analyses, it is evident that all power-related variables significantly correlate with reactive agility in boys. The correlation coefficients ranged from 0.28 to 0.51 for jumping performance, 0.62 to 0.64 for sprinting, and 0.66 to 0.74 for TCODS and 20Y performances. However, multivariate analysis revealed TCODS as the only significant predictor of TRAG. Thus, we can confirm that TCODS in the here-studied boys was an indicator of “overall motor status”. Actually, this is in accordance with previous studies where authors examined predictors of RAG in competitive athletes, where significant correlations between the sport-specific CODS and RAG performances were reported of professional futsal players, young soccer players, young tennis players, and rugby league professional players [25,26,27]. Another important element additionally explains the importance of TCODS in predicting TRAG. In brief, TCODS and TRAG had similar scenarios and consisted of similar movement patterns (please see Methods for details). While strong correlations between pre-planned and non-planned agility tests with the same movement patterns were well documented in previous studies, we have no doubt that it additionally contributed to the finding that TCODS was the only significant predictor of TRAG in this study [27,28,29].

While TCODS was a significant predictor of TRAG in girls as well, we have no doubts that the background of its influence on TRAG for girls is almost certainly very similar to the one previously discussed for boys. However, the indicators of lower body power (e.g., CMJ and DJ) were also significantly multivariately associated with TRAG among girls. The explanation of these associations should be found in the characteristics of the CMJ and DJ.

These two types of jumping are characterized by slow (CMJ) or fast (DJ) short-stretching cycles, during which, the muscle goes through the phases of eccentric, isometric, and, finally, concentric contraction [30]. In that context, the finding of significant influence on TRAG is not surprising as the same pattern of different types of muscle contraction is characteristic of agile stop-and-go movements, distinct for TRAG [31,32]. In particular, when performing such a movement, sudden deceleration with eccentric muscle contraction occurs first. After that, there is a short period of isometric contraction when the movement is stopped and, finally, concentric contraction occurs in the acceleration phase [3].

### 4.3. Gender Differences in the Prediction of the Reactive Agility Performance

From the perspective of our study, it is essential to discuss the differences in the prediction of TRAG between the genders. Specifically, lower body power significantly predicted TRAG among girls but not among boys. There are two possible explanations for such findings. The first one is “contextual” (i.e., differences in sports involvement between genders), while the second explanation is related to differences in the maturation process between boys and girls at that age.

In early puberty, boys are more involved in sports, specifically team sports that are saturated with agile movements [33,34,35]. For example, a study on a large sample of Australian adolescents from 12 to 16 years old, found that 78.5% of boys participate in organized sports compared to 66.1% of girls [35]. This is not only related to organized participation in competitive sports but also to “free play”, where boys more often than girls participate in different team sports [36,37]. Consequently, it is reasonable to expect that the boys in our sample have a higher level of specific motor skills which are (systematically and non-systematically) developed throughout participation in team sports [38]. It will help them in agility tests structured as in this study (TCODS and TRAG had the same movement patterns). On the other hand, girls (who are not as engaged in sports as boys, and therefore are relatively less skilled than boys of the same age) will probably conduct TRAG while exploiting their power capacities.

Second, the differences in biological maturity can potentially have a significant role in our findings regarding gender differences in predictors of TRAG. Namely, it is known that girls mature earlier and enter accelerated growth and development phases before boys [39,40]. Consequently, differences in power capacities such as jumping and sprinting among girls are greater than among boys, i.e., they have a greater variance in power than boys. As a result, stronger girls exploited their capacities even in RAG.

Indeed, studies have shown a more significant influence of physical capacity on agility in relatively older and more mature participants [41,42]. For example, in the study on youth football players, Krolo et al. analyzed predictors of sport-specific agility [42]. The results showed that the observed predictors, i.e., sprinting and power capacities, explained the larger percentage of agility variance in older than in younger participants [42]. Additionally, a study on pubertal handball male players showed that in older players (post-peak height velocity (PHV) group), a more considerable proportion of handball-specific agility was explained with physical capacities compared to the pre-PHV group [41]. It was explained by the fact that early maturers experienced more dynamic morphological changes and were able to generate more force than their late-maturing peers [41]. It is also important to note that changes also occur in the cognitive aspect of maturation with neural adaptations, which are an essential part of reactive agility. This not only explains our findings but also directs future studies to include cognitive parameters as agility predictors. Supportively, recent studies undertaken in other sports highlighted the applicability of the Stroop test (i.e., a test that measures the delay in reaction time between congruent and incongruent stimuli) as an important determinant of various facets of success in sports, indicating the potential usefulness of such measurement tools in determining the predictors of RAG as well [43].

### 4.4. Predictors of Reactive Agility in Children in Comparison to Predictors of Reactive Agility in Athletes

When observing all previously discussed associations between predictors and RAG, and comparing them with previous reports on athletes, certain differences in correlations should be highlighted. First, previous studies performed with athletes reported RAG as being more influenced by sprinting and jumping capacities than we have found herein. Second, the correlation between CODS and RAG in children was evidently higher than the correlation between CODS and RAG in athletes. With regard to the objectives of our study, these issues are specifically discussed.

In our study (specifically for boys), sprinting and jumping were not multivariately associated with RAG, which was not the case in previous studies performed with athletes [7,42,44] However, this is at least partially a consequence of the selection of variables in our multivariate regression. Namely, in previous studies, anthropometric indices, sprinting and jumping capacities were most often analyzed separately for both RAG and CODS, while CODS tests were not involved in the analyses as predictors of RAG [7,42,44]. For example, a study with young soccer players highlighted power capacities, manifested through slow and fast short-stretching cycles as the factors contributing to RAG [42]. Additionally, a study on a sample of top-level futsal players evidenced anthropometric indices and reactive strength as predictors of performance on the futsal-specific reactive agility test [44]. However, as we said previously, CODS was not included as a predictor of RAG in these studies, which naturally increased the percentage of the variance that was explained by other observed characteristics and capacities. However, we must not ignore the fact that RAG and CODS are more correlated in the here-studied children than in athletes observed previously, and this will also be shortly discussed.

Indeed, the correlations between the same-scenario CODS and RAG in our study are much higher (0.63 for girls and 0.74 for boys) than the correlations between the same capacities in competitive athletes. For example, Sheppard et al. (2006) reported less than 10% of the shared variance between sport-specific CODS and RAG in Australian football players, while Scanlan evidenced a negligible correlation between RAG and CODS in basketball players [45,46]. In explaining such relatively small correlations between CODS and RAG in professional athletes, authors regularly concluded that RAG performance in professional, highly trained athletes is more influenced by perceptual and cognitive abilities than by athletic parameters (i.e., anthropometric/body built indices and conditioning capacities), which are known to be determinants of CODS [45,46]. This is mostly explained by the fact that highly trained athletes have already reached a high level of conditioning status throughout systematic training, and/or sport-selection process, while perceptual and cognitive capacities are mostly “inherited” and/or at least are not systematically and specifically trained throughout sports training. Our study indirectly supports such considerations. In brief, it seems that RAG is more influenced by basic motor abilities in children than in professional athletes, at least partially due to the greater variance of these abilities in the relatively untrained population compared to highly trained athletes involved in professional sports.

### 4.5. Limitations and Strengths of the Study

One of the study’s limitations is the cross-sectional design. Therefore, a longitudinal approach and interventions are needed in future studies to obtain a clearer picture of the relations between the observed capacities. Additionally, we evidenced only a limited number of variables while not including some theoretically significant predictors of RAG (i.e., strength, flexibility, and cognitive and perceptive parameters). Finally, the sample of participants in this study was heterogenous; it included boys and girls from different sports. Thus, in the future, it is recommended to analyze agility predictors only on children that do not participate in agility-saturated sports.

To the best of our knowledge, this is the first study to evaluate the predictors of RAG among early pubescent boys and girls while evaluating the evidently important factors of RAG performance. Knowing the importance of RAG in competitive sports, we hope that our results will initiate further research.

## 5. Conclusions

Although the results of the correlation analysis showed a significant and relatively high association between all the observed motor parameters and RAG, multivariate analysis extracted CODS in both genders and sprinting/jumping among girls as the most significant predictors. Anthropometric indices were not factors of influence on RAG, which is most likely a consequence of the short duration of the RAG test applied herein and the participants’ age (pre-pubescent children).

High correlations between CODS and RAG and a relatively high proportion of the explained variance of RAG indicate that RAG in this age group is probably more related to motor abilities than cognitive factors. However, it is clear that RAG should be observed as a complex, multifactorial ability. Therefore, future studies must include other abilities that could influence agility performance, primarily cognitive, perceptual, and decision-making parameters. Finally, our findings suggest that pre-pubescent and early pubescent children should be systematically trained on basic motor abilities to achieve fundamentals for further developing RAG.

## Figures and Tables

**Figure 1 children-09-01780-f001:**
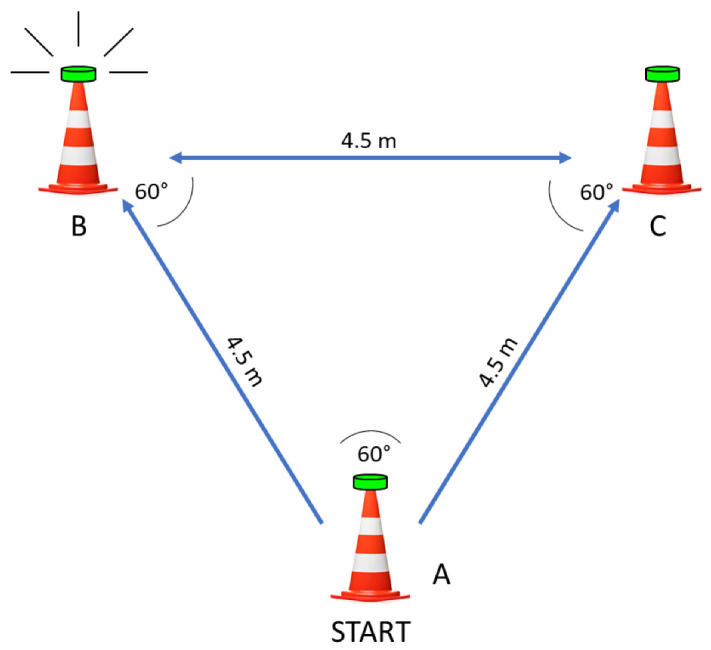
Scheme of the TRAG and TCODS testing.

**Table 1 children-09-01780-t001:** Pearson’s moment correlation coefficients between the studied variables for boys.

Var	AGE	BH	SH	BM	BF	BJ	S10	S20	20Y	TCODS	T RAG	SJ	CMJ	DJ
BH	0.36 *													
SH	0.41 *	0.66 *												
BM	0.18	0.66 *	0.53 *											
BF	−0.26 *	0.06	0.06	0.64 *										
BJ	0.36 *	0.18	0.20	−0.11	−0.52 *									
S10	−0.22	0.03	−0.02	0.34 *	0.54 *	−0.73 *								
S20	−0.21	0.03	−0.02	0.34 *	0.53 *	−0.74 *	0.98 *							
20Y	−0.26 *	−0.05	−0.02	0.23	0.42 *	−0.62 *	0.73 *	0.77 *						
TCODS	−0.22	−0.08	−0.12	0.23	0.47 *	−0.67 *	0.79 *	0.82 *	0.82 *					
TRAG	−0.18	−0.00	−0.07	0.21	0.27 *	−0.51 *	0.62 *	0.64 *	0.66 *	0.74 *				
SJ	0.27 *	0.20	0.26 *	−0.07	−0.50 *	0.78 *	−0.67 *	−0.67 *	−0.49 *	−0.56 *	−0.28 *			
CMJ	0.34 *	0.12	0.22	−0.18	−0.54 *	0.83 *	−0.73 *	−0.73 *	−0.51 *	−0.60 *	−0.35 *	0.92 *		
DJ	0.36 *	0.18	0.23	−0.10	−0.45 *	0.83 *	−0.81 *	−0.81 *	−0.55 *	−0.64 *	−0.46 *	0.87 *	0.91 *	
RSI	0.09	−0.00	0.04	−0.29 *	−0.50 *	0.69 *	−0.70 *	−0.68 *	−0.46 *	−0.54 *	−0.43 *	0.66 *	0.72 *	0.72 *

Legend: BH—body height, SH—seated height, BM—body mass, BF—body fat, BJ—broad jump, S10—10 m sprint, S20—20 m sprint, 20Y—20-yard shuttle agility test, TCODS—“Triangle” change of direction, TRAG—“Triangle” reactive agility, SJ—squat jump, CMJ—countermovement jump, DJ—drop jump, RSI—reactive strength index, * indicates the statistical significance of *p* < 0.05.

**Table 2 children-09-01780-t002:** Pearson’s moment correlation coefficients between the studied variables for girls.

Var	AGE	BH	SH	BM	BF	BJ	S10	S20	20Y	TCODS	T RAG	SJ	CMJ	DJ
BH	0.19													
SH	0.38 *	0.79 *												
BM	0.23	0.59 *	0.57 *											
BF	0.15	0.19	0.25	0.81 *										
BJ	0.09	0.37 *	0.38 *	−0.11	−0.39 *									
S10	0.01	−0.14	−0.21	0.29	0.50 *	−0.74 *								
S20	0.11	−0.13	−0.17	0.28	0.52 *	−0.70 *	0.89 *							
20Y	−0.23	−0.43 *	−0.38 *	−0.02	0.24	−0.54 *	0.51 *	0.49 *						
TCODS	−0.20	−0.08	−0.30	0.14	0.33 *	−0.43 *	0.49 *	0.44 *	0.42 *					
TRAG	0.12	0.05	−0.13	0.13	0.27	−0.33 *	0.43 *	0.39 *	0.28	0.63 *				
SJ	0.04	−0.05	0.17	−0.27	−0.46 *	0.46 *	−0.50 *	−0.51 *	−0.38 *	−0.51 *	−0.50 *			
CMJ	0.02	−0.13	0.14	−0.22	−0.36 *	0.45 *	−0.54 *	−0.56 *	−0.41 *	−0.47 *	−0.54 *	0.93 *		
DJ	0.03	−0.01	0.19	−0.27	−0.41 *	0.46 *	−0.53 *	−0.54 *	−0.42 *	−0.50 *	−0.41 *	0.89 *	0.90 *	
RSI	0.00	0.01	0.08	−0.33 *	−0.47 *	0.48 *	−0.51 *	−0.59 *	−0.45 *	−0.45 *	−0.36 *	0.75 *	0.75 *	0.86 *

Legend: BH—body height, SH—seated height, BM—body mass, BF—body fat, BJ—broad jump, S10—10 m sprint, S20—20 m sprint, 20Y—20-yard shuttle agility test, CODS—“Triangle” change of direction, TRAG—“Triangle” reactive agility, SJ—squat jump, CMJ—countermovement jump, DJ—drop jump, RSI—reactive strength index, * indicates the statistical significance of *p* < 0.05.

**Table 3 children-09-01780-t003:** Results of the multiple regression analysis for boys for TRAG as a criterion.

Predictor	β	SE of β	b	SE of b	t (64)	*p*-Value
TCODS	0.59	0.18	0.63	0.19	3.4	0.001
R = 0.80; R^2^ = 0.64; Adjusted R^2^ = 0.56; F (10.50) = 8.9; *p* < 0.001; St. Error of estimate: 0.35						

Legend: TCODS—“Triangle” change of direction speed, R—coefficient of the multiple correlation; R^2^—coefficient of the determination; β—standardized regression coefficient; b—nonstandardized regression coefficient.

**Table 4 children-09-01780-t004:** Results of the multiple regression analysis for girls with TRAG as a criterion.

Predictor	β	SE of β	b	SE of b	t (50)	*p*-Value
TCODS	0.66	0.13	0.97	0.20	4.9	0.001
CMJ	−0.86	0.31	−0.08	0.03	−2.8	0.01
DJ	−0.71	0.28	0.07	0.03	2.5	0.01
R = 0.76; R^2^ = 0.59; Adjusted R^2^ = 0.52; F (8.49) = 8.81; *p* < 0.001; St. Error of estimate: 0.39						

Legend: TCODS—“Triangle” change of direction speed, CMJ—countermovement jump, DJ—drop jump, R—coefficient of the multiple correlation; R^2^—coefficient of the determination; β—standardized regression coefficient; b—nonstandardized regression coefficient.

## Data Availability

The authors will provide data to all interested parties upon reasonable request.

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
