# Peer review of "Predictors of Reactive Agility in Early Puberty: A Multiple Regression Gender-Stratified Study"

_children, 2022, doi:10.3390/children9111780_

Round 1
Reviewer 1 Report
My recommendations are the following:
In abstract
I recommend you to describe descriptively what TCODS represents.
I recommend you to explain the context of targeting agility saturated sports, it is not clear.
Lines 22-24 mention the short duration, but previously you do not clarify the training period.
Lines 24-26 mention the basic motor skills and in the present study you will only refer to the athletic ones, I recommend the clarification.
Key words - mention power, it does not appear from the abstract that this word represents an essence of the study. I recommend the correction.
Introduction
Lines 37-39 mention two basic types of agility and descriptive use I and ii, I recommend correcting with 1 and 2.
Lines 74-80 the phrase is too generalized. The hypothesis is not clear. The purpose of the study is a finding, you have no intervention in the physical preparation of the subjects, but in the abstract you mention the preparation time. It is unclear, I recommend clarification and rewriting.
Line 83 you mentioned - aged 11-11 years, I recommend clarification. In the Participants section, do not mention the inclusion and exclusion criteria.
Lines 120-124 mention that a pretest was carried out with 21 subjects. You don't talk about this aspect in the Participants section, I recommend redoing the participant section. I recommend that the obtained results be moved to the Results section.
Lines 136-137, I think you refer to the preparation of the body for effort. Before the testing, the participants completed a 10-min , is not specific to physical activities. I recommend the correction.
I also recommend that section 2.2 be reorganized into two subsections. This section is not very clear and well organized. All information is together. This section does not explain how the TRAG is performed, which is a vital test of the study.
I recommend that in the Results section you also present descriptively their interpretation, without duplicating the information with that mentioned in the tables.
Lines 189-191 is too general and unfocused. I recommend the rewrite.
I recommend that the discussion section be expanded.
The idea is interesting, but the method of disseminating the results, the procedure and the discussions should be reconsidered. I recommend that this article be completely reorganized and reedited before being sent for publication.
Author Response
My recommendations are the following:
In abstract
I recommend you to describe descriptively what TCODS represents.
RESPONSE: Thank you for your suggestion. We modified the text and explained TCODS. Text reads: “The criterion variable was originally developed “Triangle” test of reactive agility (Triangle-RAG). Predictors included anthropometric/body composition indices (body height, seated height, body mass, and body fat percentage) and motor abilities (10 and 20 meters sprint, broad jump, squat jump, countermovement jump, and drop jump, and two tests of change of direction speed - CODS (Triangle-CODS, and 20 yards).”
I recommend you to explain the context of targeting agility saturated sports, it is not clear.
RESPONSE: Indeed, we missed to target the specific sports. It is amended now and reads: “Differences in multivariate results between genders can be explained by (i) greater involvement in agility saturated sports (i.e. basketball, tennis, soccer)” Thank you
Lines 22-24 mention the short duration, but previously you do not clarify the training period.
RESPONSE: Thank you, the details of test duration are added and text now reads: “The lack of association between anthropometric/body built and TRAG was influenced by short duration of the test (TRAG: 3.54±0.4 s).”
Lines 24-26 mention the basic motor skills and in the present study you will only refer to the athletic ones, I recommend the clarification.
RESPONSE: Thank you for your suggestion, the text is amended and now reads: “However, since in this study we practically observed only athletic abilities and anthropometric/body composition indices as predictors, in future studies other motor abilities, as well as cognitive, perceptual, and decision-making parameters as potential predictors of RAG have to be observed.”
Key words - mention power, it does not appear from the abstract that this word represents an essence of the study. I recommend the correction.
RESPONSE: Amended accordingly.
Introduction
Lines 37-39 mention two basic types of agility and descriptive use I and ii, I recommend correcting with 1 and 2.
RESPONSE: Amended as you suggested, and text reads: “In general, the existence of two basic types of agility is widely accepted and scientifically proven; (1) change of direction speed or pre-planned, non-reactive agility (CODS) and (2) reactive, non-planned agility (RAG)” (please see highlighted text – 1st paragraph of the Introduction)
Lines 74-80 the phrase is too generalized. The hypothesis is not clear. The purpose of the study is a finding, you have no intervention in the physical preparation of the subjects, but in the abstract you mention the preparation time. It is unclear, I recommend clarification and rewriting.
RESPONSE: As you suggested, the last paragraph was systematically rewritten and we hope that it is now clearer. Text reads: “However, the studies mentioned above analyzed predictors of non-reactive agility, and there is an evident lack of studies exploring predictors of RAG among children. Meanwhile, RAG is known to be an important determinant of success in agility-saturated sports [6-8,16]. Better understanding of the background of RAG in children will hopefully result in a more accurate orientation of talented children toward agility-saturated sports (i.e. basketball, soccer, handball, tennis). Therefore, the main aim of this study was to determine the association between anthropometric/body composition indices, motor abilities (predictors), and RAG in early pubescent boys and girls. We hypothesized that studied predictors would be significantly associated with RAG in studied children. Knowing the differences in fitness status between prepubescent boys and girls, we tried to avoid potential influence of gender as a covariate of established associations, and therefore gender-stratified approach was applied. “ (please see last paragraph of the Introduction)
Line 83 you mentioned - aged 11-11 years, I recommend clarification. In the Participants section, do not mention the inclusion and exclusion criteria.
RESPONSE: Thank you for noting the evident mistake (the age span was 11-12 years). Also, the inclusion/exclusion criteria are added. The text now reads: “Primary school boys (n=73) and girls (n=59) aged 11-13 years were involved in this study. In the first phase of the study the sub-sample consisting of 21 participants was tested on newly developed tests (see later for details) throughout test-retest procedure in order to evaluate reliability of the tests. All participants were of good health and were regularly attending physical education classes (PE). The inclusion criteria were: no evident motor aberrations and health related issues (as indicated by school medical staff), no locomotor injury over the period of two-weeks before testing, regular participation in PE. Exclusion criteria were: recent musculoskeletal disorders, sickness over the previous two weeks, current prevalence of pain, and/or overall sense of weakness.” (please see 1st para of the Methods section – subsection Participants)
Lines 120-124 mention that a pretest was carried out with 21 subjects. You don't talk about this aspect in the Participants section, I recommend redoing the participant section. I recommend that the obtained results be moved to the Results section.
RESPONSE: Thank you, we followed your suggestion and rewritten the participants subsection (please see previous comment and response), while results on reliability are moved to Results section (please see highlighted text at the beginning of the Results subsection).
Lines 136-137, I think you refer to the preparation of the body for effort. Before the testing, the participants completed a 10-min , is not specific to physical activities. I recommend the correction.
RESPONSE: Thank you for noticing it, it is corrected and text now reads: “Before the testing, the participants completed a 10-min warm-up including jogging, skipping, lateral running drills, dynamic stretching, and light jumping.”
I also recommend that section 2.2 be reorganized into two subsections. This section is not very clear and well organized. All information is together. This section does not explain how the TRAG is performed, which is a vital test of the study.
RESPONSE: Yes, we must agree that section on Variables was not well organized. We believe that it is now clearer. TRAG is systematically explained and scheme is included. Also, we provided video of the test execution in supplementary material.
I recommend that in the Results section you also present descriptively their interpretation, without duplicating the information with that mentioned in the tables.
RESPONSE: We tried to be more specific in results section and to avoid iteration of the data presented int Tables.
Lines 189-191 is too general and unfocused. I recommend the rewrite.
RESPONSE: Following your suggestion we tried to be more specific. The text is rewritten and now reads: This study aimed to identify predictors of reactive agility among early pubescent boys and girls. There are several most important findings. First, anthropometric/body built indices were not correlated with TRAG in studied children. Second, multivariate analysis evidenced TCODS as the only significant multivariate predictor of TRAG in boys. Mean-while, in girls, in addition to TCODS, leg power was highlighted as a significant multi-variate predictor. Therefore, our initial study hypothesis was confirmed. “ (please see 1st para of the Discussion)
I recommend that the discussion section be expanded.
RESPONSE: In this version of the manuscript the discussion is systematically rewritten in some parts (please see track changes version of the manuscript which is submitted as the supplementary material). Also, discussion section is significantly longer than in original version (previously 1600 words, now 2100 words), as a result of adding new part of the discussion (new subheading 4.4. Predictors of reactive agility in children in comparison to predictors of reactive agility in athletes).
The idea is interesting, but the method of disseminating the results, the procedure and the discussions should be reconsidered. I recommend that this article be completely reorganized and reedited before being sent for publication.
RESPONSE: Thank you for your support, and especially thank you for your valuable comments. We tried to follow it specifically, and believe that the amendments we made significantly improved the quality of the manuscript.
Staying at your disposal!
Authors
Reviewer 2 Report
Please take into account the various remarks or comments made on your manuscript. The remarks are mentioned in the file attached to this review:
Comments to authors on the manuscript.
The manuscript is written according to rigorous scientific standards.
However, the theme of the manuscript emphasizes reactive agility (RAG) as an important factor for success in sport.
The idea is built on the fact that there are no studies that have looked at reactive agility in children; hence the main aim of this study was to identify the predictors of reactive agility in boys and girls in early puberty.
The document is well written. The topic is well presented, easy to understand and the problematic is clear.
Excellent review of the literature presented on agility which is often defined as the ability to organize a movement taking into account the outcome of the motor response to be acquired. This action requires the organization of a pre-established motor pattern. Conversely, the young athlete is sometimes required to modify his or her response quickly, and in this case, the strategy adopted requires unplanned reactive agility.
According to the authors of the manuscript, this agility depends on several factors, notably morphological, anthropometric, physical and mental (involving cognitive and psychomotor perceptions of the task).
The authors proposed as their main objective to determine the association between anthropometric indices, motor skills and agility in boys and girls at the onset of puberty, hypothesizing that the predictors studied would be significantly associated with children's reactive agility, with some gender specificity.
General comments about all the manuscript:
1. P2-Line 83: replace [Primary school boys (n=73) and girls (n=59) aged 11-11 years] by [Primary school boys (n=73) and girls (n=59) aged 11-12 years were involved].
2. P3-Line 117: [5yd (4.57m)] why in yard, stay consistent. Use the international measurement unit (m).
3. P6-Line 223-228: This sentence poses a methodological bias on the interpretation of your results since you did not classify the pubertal status of your subjects according to Tanner's pubertal stage. You have considered your population to be homogeneous in age, so do not discuss your results by arguing that there are older subjects than others.
4. P7-Line 272-273: What do you mean by this sentence? [On the other hand, girls will probably conduct it more through the manifestation of power].
5. P7-Line292-293: What type of cognitive test can you propose to assess the relationship between agility and cognitive function? [This directs future studies to include cognitive parameters as agility 292 predictors].
6. P7-Line293-295: Do you have the possibility to include in your results the pubertal status of the 73 boys and 59 girls according to Tanner stages?
[Although the sample of observed participants in this study comes from the same age group, mentioned differences in biological maturity can be decisive in manifestation of different motor abilities, such as reactive agility].
7. P8-Line305-306: What kind of additional studies do you propose? Be more specific in your suggestions. [However, further studies with data on perceptual and decision factors as potential correlates of RAG in children analyses are needed for a more elaborated conclusion].
8. P8-Line312-314: In the limitations section of the study, you mention that the participants in this study were heterogeneous; it included boys and girls from different sports. In the materials and methods section, you do not mention that the participants were active in a sports club and that they belonged to different sports specialties’. [Finally, the sample of participants in this study was heterogeneous; it included boys and girls from different sports. Thus, in the future, it is recommended to analyze agility predictors only on the children that do not participate in agility-saturated sports].
9. P9-Line360: Write the reference 7 in lower case [7. Veršić, S.; Foretić, N.; Spasić, M.; Uljević, O. PREDICTORS OF AGILITY IN YOUTH SOCCER PLAYERS: CONTEXTUALIZING THE INFLUENCE OF BIOLOGICAL MATURITY. Kinesiologia Slovenica 2020, 26, 31-47.]
10. Please respond to the various comments and remarks retained after reading your manuscript.
Author Response
Please take into account the various remarks or comments made on your manuscript. The remarks are mentioned in the file attached to this review:
Comments to authors on the manuscript.
The manuscript is written according to rigorous scientific standards. However, the theme of the manuscript emphasizes reactive agility (RAG) as an important factor for success in sport. The idea is built on the fact that there are no studies that have looked at reactive agility in children; hence the main aim of this study was to identify the predictors of reactive agility in boys and girls in early puberty. The document is well written. The topic is well presented, easy to understand and the problematic is clear.
Excellent review of the literature presented on agility which is often defined as the ability to organize a movement taking into account the outcome of the motor response to be acquired. This action requires the organization of a pre-established motor pattern. Conversely, the young athlete is sometimes required to modify his or her response quickly, and in this case, the strategy adopted requires unplanned reactive agility.
According to the authors of the manuscript, this agility depends on several factors, notably morphological, anthropometric, physical and mental (involving cognitive and psychomotor perceptions of the task).
The authors proposed as their main objective to determine the association between anthropometric indices, motor skills and agility in boys and girls at the onset of puberty, hypothesizing that the predictors studied would be significantly associated with children's reactive agility, with some gender specificity.
RESPONSE: Thank you for you general positive overview of the article and for recognizing the importance of the topic. Also, we are particularly grateful for your comments and suggestions. We tried to follow it strictly and amended the manuscript accordingly. Please see following text for responses and amendments.
General comments about all the manuscript:
P2-Line 83: replace [Primary school boys (n=73) and girls (n=59) aged 11-11 years] by [Primary school boys (n=73) and girls (n=59) aged 11-12 years were involved].
RESPONSE: Thank you! Corrected
P3-Line 117: [5yd (4.57m)] why in yard, stay consistent. Use the international measurement unit (m).
RESPONSE: Text is amended and now reads: “For the 20Y test, three 50cm cones, with lighting pods on top, were positioned along a line 4.57 m (5yd) apart.”
P6-Line 223-228: This sentence poses a methodological bias on the interpretation of your results since you did not classify the pubertal status of your subjects according to Tanner's pubertal stage. You have considered your population to be homogeneous in age, so do not discuss your results by arguing that there are older subjects than others.
RESPONSE: As you suggested, we avoided discussion on maturity status as a potential factor of influence on TRAG. Thank you.
P7-Line 272-273: What do you mean by this sentence? [On the other hand, girls will probably conduct it more through the manifestation of power].
RESPONSE: The sentence is amended and we hope it is clearer. Text reads: “On the other hand, girls (who are not as engaged in sports as boys, and therefore are relatively less skilled than boys of the same age) will probably conduct TRAG while exploiting their power-capacities.”
P7-Line292-293: What type of cognitive test can you propose to assess the relationship between agility and cognitive function? [This directs future studies to include cognitive parameters as agility 292 predictors].
RESPONSE: In this version of the manuscript we shortly presented the idea of possible cingitive tests, and text reads: “Supportively, recent studies done in other sports highlighted applicability of Stroop-test (i.e. test which measures the delay in reaction time between congruent and incongruent stimuli) as important determinant of various facets of success in sport, indicating potential usefulness of such measurement tools in determining the predictors of RAG as well [37]” (please see 4th paragraph of the subsection 4.3).
P7-Line293-295: Do you have the possibility to include in your results the pubertal status of the 73 boys and 59 girls according to Tanner stages? [Although the sample of observed participants in this study comes from the same age group, mentioned differences in biological maturity can be decisive in manifestation of different motor abilities, such as reactive agility].
RESPONSE: Unfortunately, we were not able to test the children on Tanner stages due to ethical reasons. We will certainly pay attention on possible determination of biological maturity in further studies.
P8-Line305-306: What kind of additional studies do you propose? Be more specific in your suggestions. [However, further studies with data on perceptual and decision factors as potential correlates of RAG in children analyses are needed for a more elaborated conclusion].
RESPONSE: The 1st reviewer suggested changes in Discussion section, so in this version of the manuscript the part of the text you are referring is not included in the Discussion. However, directions for future studies is included in Conclusion and text reads: “Therefore, future studies must include other abilities that could influence agility performance, primarily cognitive, perceptual, and decision-making parameters. However, our findings suggest that pre-pubescent and early pubescent children should be systematically trained on basic motor abilities to achieve fundamentals for further developing RAG.” (please see highlighted text in Conclusion)
P8-Line312-314: In the limitations section of the study, you mention that the participants in this study were heterogeneous; it included boys and girls from different sports. In the materials and methods section, you do not mention that the participants were active in a sports club and that they belonged to different sports specialties’. [Finally, the sample of participants in this study was heterogeneous; it included boys and girls from different sports. Thus, in the future, it is recommended to analyze agility predictors only on the children that do not participate in agility-saturated sports].
RESPONSE: Thank you for your suggestion. Additional details about participants are now included in the Methods section, and text reads: “Primary school boys (n=73) and girls (n=59) aged 11-12 years were involved in this study. In the first phase of the study the sub-sample consisting of 21 participants was tested on newly developed tests (see later for details) throughout test-retest procedure in order to evaluate reliability of the tests. All participants were of good health and were reg-ularly attending physical education classes (PE), while some of them were included in out-of-school sports. The inclusion criteria were: no evident motor aberrations and health related issues (as indicated by school medical staff), no locomotor injury over the period of two-weeks before testing, regular participation in PE. Exclusion criteria were: recent musculoskeletal disorders, sickness over the previous two weeks, current prevalence of pain, and/or overall sense of weakness.”
P9-Line360: Write the reference 7 in lower case [7. Veršić, S.; Foretić, N.; Spasić, M.; Uljević, O. PREDICTORS OF AGILITY IN YOUTH SOCCER PLAYERS: CONTEXTUALIZING THE INFLUENCE OF BIOLOGICAL MATURITY. Kinesiologia Slovenica 2020, 26, 31-47.]7.
RESPONSE: Thank you. Corrected!
Please respond to the various comments and remarks retained after reading your manuscript.
RESPONSE: Thank you once again for your valuable comments. We tried to follow it and amended the manuscript accordingly.
Staying at your disposal!
Authors
Round 2
Reviewer 1 Report
No comments
Author Response
Thank you for recognizing the quality of our improvements!